# A Normative Theory for Causal Inference and Bayes Factor Computation in Neural Circuits

**Wen-Hao Zhang**[1,2]**, Si Wu**[3]**, Brent Doiron**[2]**, Tai Sing Lee**[1]

wenhao.zhang@pitt.edu; siwu@pku.edu.cn; bdoiron@pitt.edu; tai@cnbc.cmu.edu

[1]Center for the Neural Basis of Cognition, Carnegie Mellon University.
[2]Department of Mathematics, University of Pittsburgh.
[3]School of Electronics Engineering & Computer Science, IDG/McGovern
Institute for Brain Research, Peking-Tsinghua Center for Life Sciences, Peking University.

## Abstract

This study provides a normative theory for how Bayesian causal inference can be implemented in neural circuits. In both cognitive processes such as causal reasoning and perceptual inference such as cue integration, the nervous systems need to choose different models representing the underlying causal structures when making inferences on external stimuli. In multisensory processing, for example, the nervous system has to choose whether to integrate or segregate inputs from different sensory modalities to infer the sensory stimuli, based on whether the inputs are from the same or different sources. Making this choice is a model selection problem requiring the computation of Bayes factor, the ratio of likelihoods between the integration and the segregation models. In this paper, we consider the causal inference in multisensory processing and propose a novel generative model based on neural population code that takes into account both stimulus feature and stimulus reliability in the inference. In the case of circular variables such as heading direction, our normative theory yields an analytical solution for computing the Bayes factor, with a clear geometric interpretation, which can be implemented by simple additive mechanisms with neural population code. Numerical simulation shows that the tunings of the neurons computing Bayes factor are consistent with the "opposite neurons" discovered in dorsal medial superior temporal (MSTd) and the ventral intraparietal (VIP) areas for visual-vestibular processing. This study illuminates a potential neural mechanism for causal inference in the brain.

## 1 Introduction

Numerous psychological studies have demonstrated that perception can be formulated as Bayesian inference of the underlying causes in the world that give rise to our sensations [1–6]. These causes could be the sensory variables such as heading direction and orientation of edge, but often are causal structures from which the observations are generated. In multisensory integration, as an example, when we move around the world, the optical flows we see and the vestibular signals we experience are concordant. In this case, an integration model will be selected so that multiple cues can be weighed and combined together to form a unified estimate of head direction of self-motion [7, 8]. However, when we wear a goggle to navigate in a virtual reality world while sitting on a spinning chair, the visual and the vestibular signals would be quite discordant and it would be wrong to integrate them during inference [9]. In this case, a segregation model should be selected so that each cue will remain separated and their sources can be inferred independently. The selection of these models or latent causal structures during inference is called causal inference [10, 11].

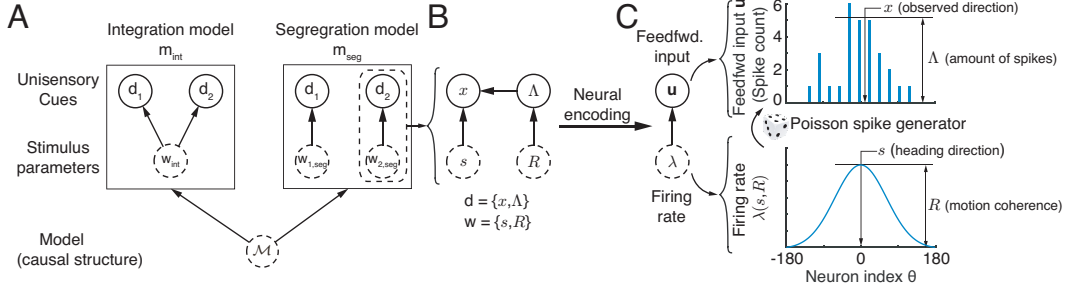

Figure 1: The generative model of causal inference. (A) The generative model. The two sensory cues are generated by the same stimulus in the integration model, while they are independently generated by two different stimuli in the segregation model. Dashed circle: latent variables; solid circle: cues (observations). (B) The likelihood function derived from neural population code. Each set of stimulus parameters include the stimulus feature $s$ (e.g., motion direction) and the stimulus strength $R$ (e.g., motion coherence). Each cue consists of observed direction $x$ and observed spike count $\Lambda$ representing the input reliability. (C) A neural encoding model where the stimulus parameters $\mathbf{w}$ and cues $\mathbf{d}$ are represented by the population firing rate $\boldsymbol{\lambda}$ and observed spiking activities $\mathbf{u}$ respectively.

A number of psychological studies have suggested that our brains indeed perform causal inference as an ideal observer (e.g., [10, 12–14]). However, it has been challenging to come up with a simple and biologically plausible neural implementation for causal inference. This is because the computation of Bayes factor, which is the ratio of likelihoods between models, requires nonlinear operations including multiplication and division, while how these nonlinear operations could be implemented by neural circuits remain a mystery [13–15]. Thus, while cue integration assuming the integration model can be accomplished by an additive mechanism through linearly summing feedforward spiking inputs in the framework of probabilistic population codes [16], a neural model with similar additive mechanisms for model selection has not been attained [15].

Here, we show that by incorporating stimulus strength or reliability $R$ (e.g. motion coherence of the visual cue, Fig. 1B) as a latent stimulus parameter to be inferred simultaneously with the stimulus feature (e.g. heading direction) in a generative model, the Bayes factor can be computed by using additive mechanism in a biologically plausible implementation. In this implementation, the neural population activities representing the Bayes factor can be computed by simply summing the inputs of one direction from one sensory modality with the inputs of the *opposite* direction from another modality. We found the tunings of these neurons representing Bayes factor in the form of neural population code are similar to the "opposite" neurons observed in MSTd and VIP whose preferred heading directions from the visual and vestibular modalities are indeed opposite, shifted by 180 degrees (Fig. 3B-C, [8, 17–19]). This work provides the first theoretical justification that the opposite cells are computing and encoding Bayes factor, which is an essential step in causal inference. We provide numerical simulation in support of this claim.

## 2 A Generative Model for Multisensory Processing

### 2.1 A probabilistic generative model

We study causal inference in the case of multisensory processing, an example of which is inferring heading direction using visual and vestibular cues [8, 19, 20]. The two cues are denoted by $\mathcal{D} = \{\mathbf{d}_l\}_{l=1}^2$, with $l = 1, 2$ representing the visual and vestibular modality respectively. Each cue can be regarded as the responses of uni-sensory neurons in visual or vestibular areas, which provide the feedforward inputs to multisensory neurons in MSTd and VIP, respectively [8, 19]. In practice, the two cues can be generated by two different models $\mathcal{M} = \{m_{\text{int}}, m_{\text{seg}}\}$, with each of them specifying an underlying causal structure (Fig. 1A, [10, 13, 14]). Each model $m_h$ ($h \in \{\text{int}, \text{seg}\}$) has its parameters $\mathcal{W}_h = \{\mathbf{w}_{lh}\}_{l=1}^2$, with $\mathbf{w}_{lh}$ denoting the parameters of stimulus of sensory modality $l$. Given a model (causal structure), the two sensory cues are generated independently (since they are generated and conveyed via different sensory pathways in the brain, Fig. 1A), i.e.,

$$p(\mathcal{D}|\mathcal{W}_h, m_h) = \prod_{l=1}^2 p(\mathbf{d}_l|\mathbf{w}_{lh}). \tag{1}$$

In the integration model $m_{\text{int}}$, there is only one source in the world, so the features of stimuli in two modalities are the same (Fig. 1A, [10, 13, 14, 21]), and we denote as $\mathbf{w}_{\text{int}} \triangleq \mathbf{w}_{1,\text{int}} = \mathbf{w}_{2,\text{int}}$. The prior of parameter $\mathbf{w}_{\text{int}}$ is assumed as a uniform distribution for simplicity,

$$p(\mathbf{w}_{\text{int}}|m_{\text{int}}) = \mathcal{U}(\mathbf{w}_{\text{int}}). \tag{2}$$

In the segregation model $m_{\text{seg}}$, there are two independent sources (Fig. 1A). Thus, the stimulus parameters in two modalities are independent with each other, and also satisfy the uniform distribution,

$$p(\mathbf{w}_{1,\text{seg}}, \mathbf{w}_{2,\text{seg}}|m_{\text{seg}}) = p(\mathbf{w}_{1,\text{seg}}|m_{\text{seg}})p(\mathbf{w}_{2,\text{seg}}|m_{\text{seg}}) = \mathcal{U}(\mathbf{w}_{1,\text{seg}})\,\mathcal{U}(\mathbf{w}_{2,\text{seg}}). \tag{3}$$

Notably, the two causal models are *mutually exclusive* to each other, in term of that only one of them holds at a single moment. The prior of the two models are assumed to be the same,

$$p(m_{\text{int}} = 1) = p(m_{\text{seg}} = 1) = 1/2. \tag{4}$$

Combing the likelihood and prior above, the whole generative process is summarized as,

$$p(\mathcal{D}, \mathcal{W}_h, m_h) = p(\mathcal{D}|\mathcal{W}_h, m_h)p(\mathcal{W}_h|m_h)p(m_h),$$
$$\propto \begin{cases} p(\mathbf{d}_1|\mathbf{w}_{\text{int}})p(\mathbf{d}_2|\mathbf{w}_{\text{int}}), & m_h = m_{\text{int}}, \\ p(\mathbf{d}_1|\mathbf{w}_{1,\text{seg}})p(\mathbf{d}_2|\mathbf{w}_{2,\text{seg}}), & m_h = m_{\text{seg}}. \end{cases} \tag{5}$$

## 2.2 Neural population code

In the framework of neural population code [16, 22], the above generative model (Eq. 5) can be described more specifically, which is a key step in linking abstract causal inference with neural circuit. Consider that $\mathbf{w}_l = \{s_l, R_l\}$ are the stimulus parameters of modality $l$, which is the heading direction ($s_l$) and its reliability ($R_l$). The stimulus information is conveyed by the responses of $N$ uni-sensory neurons in modality $l$, denoted as $\mathbf{u}_l = \{u_{lj}\}_{j=1}^N$, which satisfy the Poisson statistics (Fig. 1C, [16]),

$$p\left(\mathbf{u}_l|\lambda_{lj}(s_l, R_l)\right) = \prod_{j=1}^N \text{Poisson}(u_{lj}|\lambda_{lj}) = \prod_{j=1}^N \frac{\lambda_{lj}^{u_{lj}}}{u_{lj}!}e^{-\lambda_{lj}}, \tag{6}$$

where $\lambda_{lj}$ is the firing rate of neuron $u_{lj}$ and is a function of stimulus parameters $s_l$ and $R_l$,

$$\lambda_{lj}(s_l, R_l) = R_l \exp\left[a\cos(s_l - \theta_j) - a\right], \tag{7}$$

where $\theta_j$ is the preferred direction of neuron $u_{lj}$, and $a$ is the width of the tuning function. Here, we assume that the stimulus reliability is encoded by the peak firing rate of neurons [16, 22].

Although the neuronal responses $\mathbf{u}_l$ are high-dimensional, the likelihood function of the stimulus parameters $\mathbf{w}_l$ given $\mathbf{u}_l$ can be fully specified by two one-dimensional variables (sufficient statistics), which correspond to the readout (via population vector) of the direction ($x_l$) from $\mathbf{u}_l$ [23] and the total spike count ($\Lambda_l$),

$$x_l = \arg\left(\sum_j u_{lj}e^{i\theta_j}\right) = \tan^{-1}\left(\frac{\sum_j u_{lj}\sin\theta_j}{\sum_j u_{lj}\cos\theta_j}\right), \quad \Lambda_l = \sum_j u_{lj}. \tag{8}$$

The sufficient statistics $\mathbf{d}_l = \{x_l, \Lambda_l\}$ correspond to the sensory cues in Eq. (1). The likelihood function of stimulus parameter $\mathbf{w}_l$ derived from neural population code is calculated to be (see details in Supplementary Information (SI) 4),

$$p\left(\mathbf{d}_l = \{x_l, \Lambda_l\}|s_l, R_l\right) = \mathcal{M}\left(x_l|s_l, a\rho\Lambda_l\right)\text{Poisson}(\Lambda_l|\beta R_l),$$
$$\propto \mathcal{M}\left(s_l|x_l, a\rho\Lambda_l\right)\Gamma(R_l|\Lambda_l + 1, \beta), \tag{9}$$

where $\mathcal{M}(x)$, $\text{Poisson}(x)$ and $\Gamma(x)$ denote a von Mises, a Poisson, and a Gamma distributions, respectively. $\rho$ and $\beta$ represent the width of $\mathbf{u}_l$ and the sum of normalized firing rates, respectively (see SI. 4). The priors of $s_{lh}$ and $R_{lh}$ are assumed to be independent with each other (Eq. 2), i.e.,

$$\mathcal{U}(\mathbf{w}_{lh}) = \mathcal{U}(s_{lh})\,\mathcal{U}(R_{lh}) = (L_s L_R)^{-1}, \tag{10}$$

where $L_s$ and $L_R$ are the lengths of the spaces of $s$ and $R$, respectively. For heading direction $s$, $L_s = 2\pi$. Combining Eqs. (5, 9 and 10) together, the generative model in the form of neural population code is expressed as,

$$p(\mathcal{D}, \mathcal{W}_h, m_h) \propto \begin{cases} \prod_{l=1}^2 \mathcal{M}\left(x_l|s_{\text{int}}, a\rho\Lambda_l\right)\text{Poisson}(\Lambda_l|\beta R_{\text{int}}), & m_h = m_{\text{int}}, \\ \prod_{l=1}^2 \mathcal{M}\left(x_l|s_{l,\text{seg}}, a\rho\Lambda_l\right)\text{Poisson}(\Lambda_l|\beta R_{l,\text{seg}}), & m_h = m_{\text{seg}}. \end{cases} \tag{11}$$

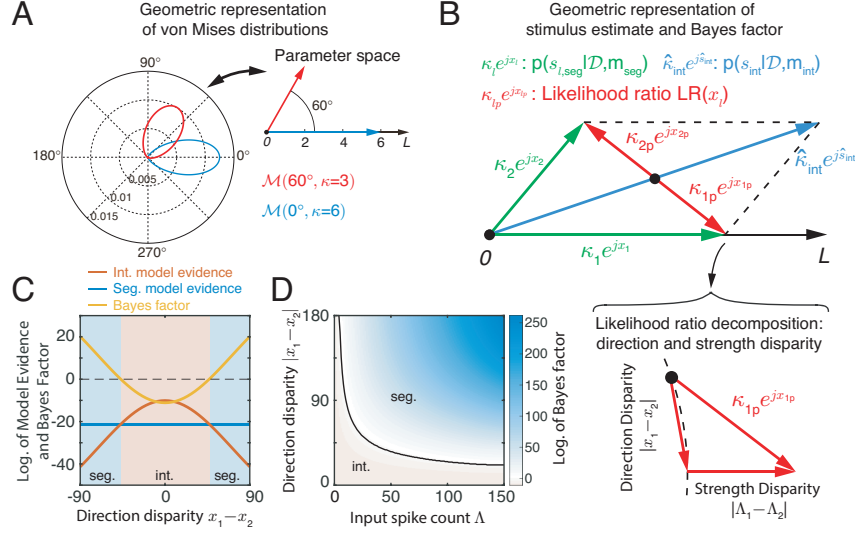

Figure 2: The geometric representation of Bayes factors. (A) The geometric representation of a von Mises distribution where its mean and concentration can be represented by the angle and length of a vector in a 2d plane respectively. (B) The geometric representation of the posterior of direction under two models (Eq. 20) and the best-fit likelihood ratios (Eq. 25) in Bayes factor. Bottom: the likelihood ratios depends on the disparity of direction as well as strength. Dashed line: the radius of half blue vector. (C) Evidence of integration and segregation models and Bayes factor with cue directions. (D) The decision boundary with input spike count, where the spike count of two cues are always the same, i.e., $\Lambda_1 = \Lambda_2$. Parameters: $L_s = 2\pi$, $L_R = 100$ Hz, $a = 3$ and $N = 180$. (C) $\Lambda_1 = \Lambda_2 = 30$.

Notably, the generative model considered in the present study (Eq. 9) includes explicitly the stimulus strength $R$, which was treated as a "nuisance" parameter in previous studies (e.g., [13–15]). We claim that it is important for the neural system to exploit the disparity of the strength $R$ of two stimuli to perform causal inference. For example, when you are watching a shaky video (low motion coherence) in virtual reality while you are walking straight ahead in real world (high reliability), even if the moving direction and the speed of optic flow in virtual reality is the same as your actual walking, you probably feel the optic flow is not generated by your walking and even feel motion-sickness because of the difference of motion coherence between visual and vestibular stimuli.

## 3 Bayesian Causal Inference

In order to interpret the world, the neural circuit needs to infer the underlying causal structure $m_h$ based on sensory cues $\mathcal{D}$ (Fig. 1), which can be achieved through estimating the posterior of each model $m_h$. Although the spike count $\Lambda_l$ is observed, the neural circuit is assumed to be only interested in the heading direction $\mathbf{x} = \{x_1, x_2\}$, and evaluates each model's feasibility by its performance in explaining $\mathbf{x}$. According to the Bayes' theorem, the posterior of the integration model $m_{\text{int}}$ is,

$$p(m_{\text{int}}|\mathbf{x}) = \frac{p(\mathbf{x}|m_{\text{int}})p(m_{\text{int}})}{\sum_h p(\mathbf{x}|m_h)p(m_h)} = \left[1 + \frac{p(\mathbf{x}|m_{\text{seg}})}{p(\mathbf{x}|m_{\text{int}})}\right]^{-1}, \tag{12}$$

where the condition that two models have the same prior is used (Eq. 4), i.e., $p(m_{\text{seg}})/p(m_{\text{int}}) = 1$. Since there are only two models, it always has $\sum_h p(m_h|\mathbf{x}) = 1$, and knowing the posterior of one model fully determines the posterior of another. From Eq. (12), we see the key of causal inference is to calculate the likelihood ratio between two models, which is called the Bayes factor [24, 25],

$$\mathcal{B}(\mathbf{x}) = \frac{p(\mathbf{x}|m_{\text{seg}})}{p(\mathbf{x}|m_{\text{int}})}. \tag{13}$$

If the Bayes factor is less than 1, $p(m_{\text{int}}|\mathbf{x}) > p(m_{\text{seg}}|\mathbf{x})$ and the integration model is favoured; otherwise the segregation model is chosen. The core of computing the Bayes factor is to evaluate

the evidence of each model $p(\mathbf{x}|m_h)$, which needs to marginalize the parameters $\mathcal{W}_h$ and the spike counts $\Lambda = \{\Lambda_1, \Lambda_2\}$,

$$p(\mathbf{x}|m_h) = \int \int p(\mathbf{x}, \mathbf{\Lambda}|\mathcal{W}_h)p(\mathcal{W}_h|m_h)d\mathcal{W}_h d\mathbf{\Lambda},$$

$$\simeq \underbrace{p(\mathbf{x}|\hat{\mathcal{W}}_h)}_{\text{Best-fit likelihood}} \times \underbrace{p(\hat{\mathcal{W}}_h|m_h)\det(\mathcal{H}_h/2\pi)^{-\frac{1}{2}}}_{\text{Occam factor, OF}(m_h)}, \tag{14}$$

where the Laplace's method is used to approximate the double integral (see SI. 2, [25, 26]), which works well when the spike counts $\Lambda$ are sufficiently large (Fig. S1, see details in SI. 5). Computing the evidence of each model needs to fit the model to explain the sensory cues. Denote $\hat{\mathcal{W}}_h = \{\hat{s}_{lh}, \hat{R}_{lh}\}_{l=1}^2$ the best-fit parameters (the maximal posterior estimate) of model $m_h$, i.e.,

$$\hat{\mathcal{W}}_h = \arg\max_{\mathcal{W}_h} p(\mathcal{W}_h|\mathcal{D}, m_h). \tag{15}$$

The best-fit likelihood of the observed direction $\mathbf{x}$ is given by (see details in SI. 5.3),

$$p(\mathbf{x}|\hat{\mathcal{W}}_h) = \prod_{l=1}^2 p(x_l|\hat{\mathbf{w}}_{lh}) \simeq \prod_{l=1}^2 \mathcal{M}(x_l|\hat{s}_{lh}, a\rho\beta\hat{R}_{lh}). \tag{16}$$

In Eq. (14), $\mathcal{H}_h = -\nabla\nabla \ln p(\hat{\mathcal{W}}_h|\mathcal{D}, m_h)$ is the negative Hessian matrix of the logarithm of the posterior $p(\mathcal{W}_h|\mathcal{D}, m_h)$, reflecting the uncertainty of the inferred parameter $\mathcal{W}_h$.

Note that causal inference (model selection) is not simply choosing a causal structure (model) which best explains the observed direction $\mathbf{x}$, since a complex model can always fit the data well. An over-parameterized model or a model requiring too much fine-tuning will be rejected, and this is captured by the Occam factor $\text{OF}(m_h)$ in Eq. (14). The Occam factor for a complex model is small, since the probability of choosing a particular parameter value $p(\hat{\mathcal{W}}_h|m_h)$ is low due to the large parameter space; and a fine-tuned model has a large $\mathcal{H}_h$, which also reduces the Occam factor [26].

In summary, Bayesian causal inference undergoes two levels of inference: the first level is inferring the best-fit parameters $\hat{s}$ and $\hat{R}$ given each model (Eq. 15); and the second level is inferring the models $\mathcal{M}$ by using the best-fit parameters to evaluate each model's performance, with the model complexity penalized by the Occam factor (Eq. 14). In the section below, we presented how the two levels of inference are performed.

## 3.1 Maximum posterior estimate of stimulus parameters

In the segregation model $m_{\text{seg}}$, each cue $\mathbf{d}_l$ is exclusively used to fit the parameters $\mathbf{w}_{lh}$ (Eq. 11),

$$p(\mathcal{W}_{\text{seg}}|\mathcal{D}, m_{\text{seg}}) \propto \prod_{l=1}^2 \mathcal{M}(s_{l,\text{seg}}|x_l, \kappa_l \triangleq a\rho\Lambda_l)\Gamma(R_{l,\text{seg}}|\Lambda_l + 1, \beta), \tag{17}$$

and the maximum-posterior estimates of the parameters are (see details in SI. 5.1),

$$\hat{s}_{l,\text{seg}} = x_l, \quad \hat{R}_{l,\text{seg}} = \Lambda_l/\beta. \tag{18}$$

On the other hand, the integration model $m_{\text{int}}$ only has one set of parameters $\mathbf{w}_{\text{int}} = \{s_{\text{int}}, R_{\text{int}}\}$ (Eq. 2), whose estimate involves combining two cues together (Eq. 9),

$$p(\mathbf{w}_{\text{int}}|\mathcal{D}, m_{\text{int}}) \propto \prod_{l=1}^2 \mathcal{M}(s_{\text{int}}|x_l, \kappa_l)\Gamma(R_{\text{int}}|\Lambda_l + 1, \beta),$$

$$\propto \mathcal{M}(s_{\text{int}}|\hat{s}_{\text{int}}, \hat{\kappa}_{\text{int}})\Gamma(R_{\text{int}}|\Lambda_1 + \Lambda_2 + 1, 2\beta). \tag{19}$$

The parameters $\hat{s}_{\text{int}}$ and $\hat{\kappa}_{\text{int}}$ of the posterior of direction satisfy [27] (see details in SI. 5.2),

$$\hat{\kappa}_{\text{int}}e^{j\hat{s}_{\text{int}}} = \kappa_1 e^{jx_1} + \kappa_2 e^{jx_2}. \tag{20}$$

Combining the above results (Eqs. 19-20), the parameter estimates in the integration model are,

$$\hat{s}_{\text{int}} = \tan^{-1}\left(\frac{\kappa_1 \sin x_1 + \kappa_2 \sin x_2}{\kappa_1 \cos x_1 + \kappa_2 \cos x_2}\right), \quad \hat{R}_{\text{int}} = \frac{\Lambda_1 + \Lambda_2}{2\beta}. \tag{21}$$

It is worthy to note that there is a clear geometric interpretation of the parameters in the posterior of direction $s$ (Eqs. 17 and 19). The parameters of a von Mises distribution $\mathcal{M}(s|x, \kappa)$ can be represented by the vector $\kappa e^{jx}$ in a two-dimensional parameter plane with its mean $x$ and concentration $\kappa$ represented by the angle and length of the vector, respectively (Fig. 2A). Thus, the posterior of direction in the segregation model ($\mathcal{M}(s_{l,\text{seg}}|x_l, \kappa_l)$ in Eq. 17) can be represented by two green vectors $\kappa_l e^{jx_l}$ in Fig. 2B. In comparison, since the integration model combines the two cues together, the posterior of direction in the integration model can be represented by the blue vector in Fig. 2B, which is the sum of the two green vectors (Eq. 20). The geometry in the parameter space shows that the integration model accumulates the *common information* of two cues to estimate stimulus, and the estimate of the integration model is always the consensus (reliability based average) of cues.

## 3.2 Occam factors of two models

The Occam factors of two models are (substituting Eqs. (18, 21) into Eq. (14), see SI. 5 for details),

$$\text{OF}(m_{\text{seg}}) = 4 \times \text{OF}(m_{\text{int}})^2, \quad \text{OF}(m_{\text{int}}) = \pi[L_s L_R \sqrt{a\rho}\beta]^{-1}. \tag{22}$$

The $\text{OF}(m_{\text{seg}})$ is smaller than $\text{OF}(m_{\text{int}})$ by a order, because the number of parameters in the segregation model is double that in the integration model. Moreover, the Occam factors of the two models are invariant constants with input spikes $\Lambda_l$ and direction $x_l$, because the dependence of the uncertainties of $s_{lh}$ and $R_{lh}$ on $\Lambda_l$ cancel, which greatly simplify the neural implementation.

## 3.3 The Bayes factor

Once the best-fit stimulus parameters (Eqs. 18 and 21) and the Occam factors (Eq. 22) are obtained, the Bayes factor determining two models can be calculated as a function of heading direction (Eq. 13),

$$\mathcal{B}(\mathbf{x}) \simeq \prod_{l=1}^{2} \frac{\mathcal{M}(x_l|\hat{s}_{l,\text{seg}}, \kappa_l)}{\mathcal{M}(x_l|\hat{s}_{\text{int}}, \hat{\kappa}_{\text{int}}/2)} \frac{\text{OF}(m_{\text{seg}})}{\text{OF}(m_{\text{int}})} = \prod_{l=1}^{2} \text{LR}(x_l) \times \text{OFR}, \tag{23}$$

where $\text{LR}(x_l)$ is the ratio of the best-fit likelihoods of two models, and $\text{OFR} = \text{OF}(m_{\text{seg}})/\text{OF}(m_{\text{int}})$ is the Occam factor ratio which is a constant invariant to input (Eq. 22). In Eq. (23), $\kappa_l = a\rho\beta\hat{R}_{l,\text{seg}}$ (Eq. 17) and $\hat{\kappa}_{\text{int}}/2 \approx (\kappa_1 + \kappa_2)/2 = a\rho\beta\hat{R}_{\text{int}}$ due to $|x_1 - x_2| \ll \min(\kappa_1, \kappa_2)$ (Eq. 20). Note that the concentration of the best-fit likelihood of the integration model, i.e., $\hat{\kappa}_{\text{int}}/2$ in the denominator of Eq. (23), is half of the concentration of the posterior, i.e., $\hat{\kappa}_{\text{int}}$ in Eqs. (19-20). Intuitively, this is due to that the integration model uses the two cues' consensus (average) to explain each cue. When the cues are from the same source, their consensus is similar with themselves statistically.

Since the Occam factor ratio OFR is a constant invariant with inputs, computing the dependency of Bayes factor on inputs lies in the computation of likelihood ratio $\text{LR}(x_l)$. Notably, the ratio between two circular distributions is still a circular distribution. Dividing by a circular distribution is proportional to rotating the distribution to opposite direction and multiplying it (comparing the below equation with Eq. 23), i.e.,

$$\text{LR}(x_l) \propto \mathcal{M}(x_l|\hat{s}_{l,\text{seg}}, \kappa_l)\mathcal{M}(x_l|\hat{s}_{\text{int}} + \pi, \hat{\kappa}_{\text{int}}/2) = A \times \mathcal{M}(x_l|x_{lp}, \kappa_{lp}), \tag{24}$$

where $A$ is the product of normalizing constants[1]. Using Eq. (20), the parameters $x_{lp}$ and $\kappa_{lp}$ of $\text{LR}(x_l)$ are calculated as,

$$\kappa_{lp}e^{jx_{lp}} = (\kappa_l e^{jx_l} - \kappa_{l'}e^{jx_{l'}})/2 = [\kappa_l e^{jx_l} + \kappa_{l'}e^{j(x_{l'}+\pi)}]/2, \quad l' = 3 - l. \tag{25}$$

Geometrically, the likelihood ratio parameters ($x_{lp}$ and $\kappa_{lp}$ in Eqs. 24-25) correspond to the difference between green vectors (the best-fit likelihood of the segregation model) and half of the blue vector (the best-fit likelihood of the integration model), and they are represented by two red vectors in Fig. 2B. This geometrical relationship suggests that the likelihood ratio takes into account the disparities of both direction $|x_1 - x_2|$ and strength $|\Lambda_1 - \Lambda_2|$ (Fig. 2B bottom), and reflects how well the integration model can explain the two cues, as the lengths of the red vectors increase with the cue disparity. From the property of parallelogram, the two red vectors are always of the same length but point to the opposite direction with each other, implying the parameters of the two likelihood ratios have the same concentration, i.e., $\kappa_{1p} = \kappa_{2p}$, but opposite means, i.e., $x_{2p} = x_{1p} + \pi$.

Fig. 2 presents the results of model evidence and Bayes factor. The evidence of the segregation model, $p(\mathbf{x}|m_{\text{seg}})$, is a constant irrelevant of $|x_1 - x_2|$ (Fig. 2C, blue line), since each cue is independently fit by a parameter and hence the cues can always be perfectly fit regardless of their disparity. However, the segregation model is penalized by the Occam factor much more compared with the integration model since it has more parameters (Eq. 22). In contrast, the integration model parsimoniously uses the two cues' consensus to explain cues, and hence its explanatory power, $p(\mathbf{x}|m_{\text{int}})$, decreases with the cue disparity (Fig. 2C, red line). In summary, the integration model will be favoured when two cues are similar (Fig. 2C), consistent with the intuition that cues from the same object will be statistically more similar than cues from different objects (Fig. 1A) [12, 13].

The spike counts $\mathbf{\Lambda}$ affects the integration probability indirectly through the estimate of $R$ (Eqs. 18 and 21). When the spike counts of both cues are low, i.e., noisy cues due to low motion coherence,

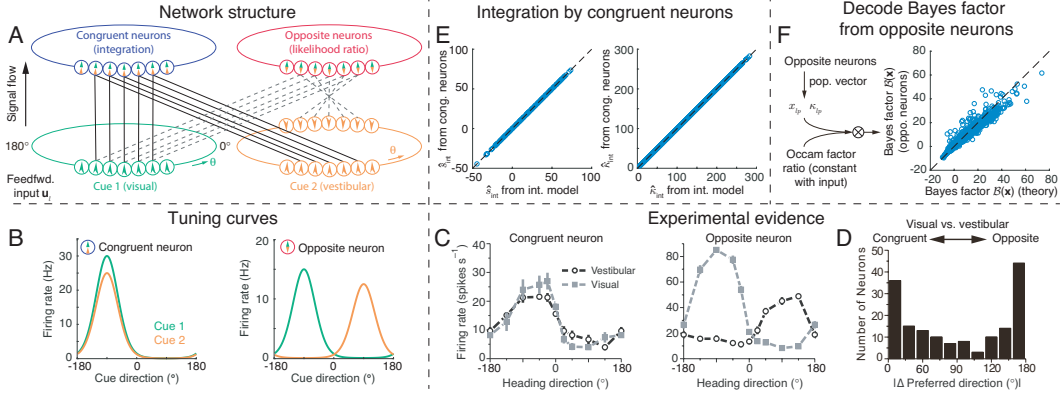

Figure 3: Congruent and opposite neurons implement the integration and the Bayes factor respectively. (A) The schematic of the network structure. Congruent (opposite) neurons receive the feedforward inputs from two cues in a congruent (opposite) manner. Each circle represents a neuron where the two arrows inside denotes its preferred directions under two sensory modalities, with the color specifying the modality. (B) The tunings of an example congruent and opposite neuron in the network given two sensory cues. For illustration, the strength of two cues is set to be different. (C) Tuning curves of a congruent and an opposite neuron in both MSTd and VIP (adapted from [8]). (D) The number of congruent and opposite neurons in MSTd and VIP (adapted from [17]). (E) The comparison between the mean $\hat{s}_{\text{int}}$ and concentration $\hat{\kappa}_{\text{int}}$ decoded from congruent neurons with the theoretical predictions (Eq. 19). (F) The Bayes factors decoded from opposite neurons are compared with theoretical prediction (Eq. 23). Parameters: (E-F) $s_1 = 0°$, $s_2 \in [0°, 20°]$, $R_l \in [5, 50]$Hz.

the system tends to integrate two cues together to increase the confidence of the stimulus estimate, so the range of integration is large (Fig. 2D). In contrast, in the case of large spike counts, the estimate of each cue is reliable enough even without integration, and the system can discriminate the disparity between two cues clearly and the range of integration shrinks.

## 4 Neural Implementation of Causal Inference

We further explore how causal inference can be implemented in neural circuits. As described above, causal inference involves two operations, estimating the best-fit stimulus parameters of each model (Eq. 15) and calculating the Bayes factor (Eq. 23). The neural system needs at least two populations of neurons to implement each of them.

### 4.1 Congruent neurons responsible for cue integration

Since the estimate of the stimulus parameters in the segregation model is the same as the likelihood (Eq. 18), the feedforward inputs $\mathbf{u}_l$ represents the estimate of the segregation model already.

The integration model combines two cues together. Following the idea of [16], cue integration can be achieved by a population of neurons which sum the feedforward inputs of two cues together (see derivations in SI. 6.1). Denote the responses of these neurons by $\mathbf{r}^c$, we have,

$$r^c(j) = u_1(\theta_j) + u_2(\theta_j), \tag{26}$$

where $u_l(\theta_j)$ denotes the input from modality $l$ with preferred direction $\theta_j$ given cue $l$ (Eq. 6). The preferred direction of $r^c(j)$ under two cues are the same (Fig. 3B), consistent with the tuning of congruent neurons found in MSTd and VIP (Fig. 3C, [8, 19]), which are known to be responsible for cue integration [8, 16, 28].

### 4.2 Opposite neurons representing the Bayes factor

The core of computing Bayes factor is the likelihood ratio (Eq. 23), because the Occam factor of two models are both constants invariant with inputs (Eq. 22). Thus we consider another population of neurons computing the likelihood ratio $\text{LR}(x_l)$ in Bayes factor. Since two likelihood ratios $\text{LR}(x_l)$

are always opposite to each other, they can be parsimoniously represented by the same population of neurons. Eqs. (24-25) reveal that the likelihood ratio is proportional to the product of two best-fit likelihoods but in the opposite manner. Analogous to the neural implementation of cue integration (Eq. 19), the ratio $\text{LR}(x_1)$ can be represented by another population of neurons averaging the two feedforward inputs in an opposite manner (see details in SI. 6.2), whose responses $\mathbf{r}^o$ are given by,

$$r^o(j) = [u_1(\theta_j) + u_2(\theta_j + \pi)]/2. \tag{27}$$

The preferred direction of $r^o(j)$ under modality 1 is $\theta_j$, but becomes $\theta_j + \pi$ under modality 2 (Fig. 3B). Experiments also found such kind of "opposite" neurons in MSTd and VIP whose preferred directions are opposite in response to visual and vestibular cues (Fig. 3C, [8, 19]). Note that the two populations of neurons explicitly represent the distributions of stimulus direction, while the estimate of stimulus strength $\hat{R}_{\text{int}}$ implicitly affects the total responses of opposite neurons as the average of the two inputs (Eq. 27), in contrast to congruent neurons which sum up two inputs (Eq. 26).

## 4.3   Simulation results

We simulate a population of congruent neurons and a population of opposite neurons with equal number, as found in the experiments (Fig. 3D, [17, 18]). The congruent neurons' responses $\mathbf{r}^c$ sum up the two feedforward inputs (two cues) together (Eq. 26), while the opposite neurons' responses $\mathbf{r}^o$ average the two feedforward inputs in an opposite manner (Eq. 27, Fig. 3A, see details in SI. 7). We decode the mean and concentration of the heading direction from the congruent neurons $\mathbf{r}^c$ via population vector (Eq. 8, [23]) and compare the results with the posterior of direction derived from theory (Eq. 20, see details in SI. 7). Meanwhile, we decode the mean and concentration from the opposite neurons $\mathbf{r}^o$ as the neurons' estimate for $x_{1p}$ and $\kappa_{1p}$, which are the parameters of $\text{LR}(x_1)$. The parameters of $\text{LR}(x_2)$ can be obtained by using the relations $x_{2p} = -x_{1p}$ and $\kappa_{2p} = \kappa_{1p}$, because the two likelihood ratios have the same length but opposite direction (Fig. 2B). The Bayes factor will be obtained by multiplying the decoded likelihood ratios from opposite neurons with the constant Occam factor ratios (Eqs. 23-24). We then compare the decoded posteriors represented by the congruent neurons, and the Bayes factor decoded from opposite neurons with theoretical predictions (Fig. 3E-F). The results confirm that the congruent neurons achieve cue integration, and the opposite neurons compute and represent the likelihood ratio in the Bayes factor.

## 5   Conclusions and Discussions

This study develops a normative theory to address how causal inference can be implemented by simple additive mechanisms in neural circuits, and demonstrate that the opposite neurons found in MSTd and VIP could compute and represent the likelihood ratios in Bayes factor in a generative model framework based on probabilistic population code. Our theory also provides a geometric interpretation of causal inference which illuminates clearly how the Bayes factor and cue integration depend on the input direction and strength. Compared to existing proposed complex neural circuits for causal inference, our model is rather simple, relying only on an additive operation, and is hence biologically more plausible. Notably, opposite neurons have been known for more than a decade, yet their precise computational and functional roles remain unclear [17, 19]. Here, our study suggests that opposite neurons are responsible for implementing causal inference in neural systems.

Previous works exploring the implementation of causal inference in neural systems (e.g., [15]) have not associated their models with the neuronal properties found in the cortex. An important insight from our study is that in computing Bayes factor, opposite neurons need to take into account not only the difference in the heading directions, but also the difference in the stimulus strength or reliability of signals, from the two sensory modalities (Fig. 2B), which is an issue missed in the previous works (e.g. [13, 15]). Previous theoretical works also suggested that opposite neurons compute the ratio between distributions [27, 29], but they consider the difference of the inferred common stimulus direction $s_{\text{int}}$ from two cues, i.e., the posterior ratio of the stimulus direction [29]. Here, we consider the opposite neurons compute the difference of the reconstructions of the input $x$ from two models, i.e., the ratio of best-fit likelihoods (Eq. 14).

Nevertheless, we would like to point out that our theory on neural computation and representation of Bayes factor in the current form only holds for circular variables, such as direction or orientation. How the Bayes factor of a non-periodic variable, e.g., depth or spatial location, are computed by neurons remains unclear. Further experimental evidence for cue integration with non-periodic variables are

needed to address this issue. Furthermore, the present study mainly focuses on the computation of Bayes factor, and how the neural system carries out the followed computations based on the inferred causal structure has yet been explored, which forms our future research.

### Acknowledgments

This work is supported by National Science Foundation (1816568), Intelligence Advanced Research Projects Activity (D16PC00007), the National Institutes of Health (Grants 1U19NS107613-01 and R01EB026953), the Vannevar Bush Faculty (Fellowship N00014-18-1-2002), and the Simons Foundation Collaboration on the Global Brain. We also thank Rob Kass and Xaq Pitkow for their useful suggestions.

## Footnotes

[1] $A = 2\pi I_0(\hat{\kappa}_{\text{int}}/2)I_0(\kappa_{lp})/I_0(\kappa_l)$. $I_0(x)$ is the modified Bessel function of the first kind and zero order.

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
