[Supplementary Material]

# A Normative Theory for Causal Inference and Bayes Factor Computation in Neural Circuits: Supplementary Information

**Wen-Hao Zhang**[1,2]**, Si Wu**[3]**, Brent Doiron**[2]**, Tai Sing Lee**[1]

`wenhao.zhang@pitt.edu; siwu@pku.edu.cn; bdoiron@pitt.edu; tai@cnbc.cmu.edu`

[1]Center for the Neural Basis of Cognition, Carnegie Mellon University.
[2]Department of Mathematics, University of Pittsburgh.
[3]School of Electronics Engineering & Computer Science, IDG/McGovern
Institute for Brain Research, Peking-Tsinghua Center for Life Sciences, Peking University.

## Contents

# 1  Supplementary Figures

Figure S1: Comparison of the model evidence calculated from numerical calculation and Laplacian approximation. (A) The likelihood function of cue direction and spike count, $p(x, \Lambda|s, R)$ in Eq. (9) (top). Marginalizing the spike count yields the likelihood for cue direction, $p(x|s, R)$ (bottom). Dots and line indicate the likelihood obtained from numerical simulation and Laplacian approximation (Eq. 16 and Sec. 5.3). (B) Comparison of the concentration of the likelihood function of cue direction obtained from numerical simulation and Laplacian approximation.

## 2   Laplacian Approximation

Laplace's method is approximating the integral of a function by using its second order expansion around a peak value. We assume that an unnormalized probability density $p^*(x)$, and we are interested in finding its normalizing constant,

$$Z_p = \int p^*(x)dx. \tag{S1}$$

The $p^*(x)$ has a peak at point $x_0$. We can take Taylor expansion of logarithm of $p^*(x)$ around $x_0$,

$$\ln p^*(x) \simeq \ln p^*(x_0) - \frac{h}{2}(x-x_0)^2 + \mathcal{O}((x-x_0)^n). \tag{S2}$$

Note that the first order expansion is zero on $x_0$ because $x_0$ is a maximum point. In above equation,

$$h = -\frac{\partial^2}{\partial x^2}\ln p^*(x)\Big|_{x=x_0}. \tag{S3}$$

We then approximate $p^*(x)$ by an unnormalized Gaussian,

$$Q^*(x) = p^*(x_0)\exp\left[-\frac{h}{2}(x-x_0)^2\right], \tag{S4}$$

and then the normalizing constant $Z_p$ can be approximated by the normalizing constant of $Q^*(x)$,

$$Z_Q = p^*(x_0)\sqrt{2\pi h^{-1}} \tag{S5}$$

For a multivariate variable $\mathbf{x}$ satisfying a unnormalized distribution $p^*(\mathbf{x})$, its normalizing constant can be also similarly approximated,

$$Z_p \simeq Z_Q = p^*(\mathbf{x}_0)\det(\mathbf{H}/2\pi)^{-1/2}, \tag{S6}$$

where $\mathbf{H}$ is the negative Hessian matrix

$$\mathbf{H}_{ij} = -\frac{\partial^2}{\partial x_i \partial x_j}\ln p^*(\mathbf{x})\Big|_{\mathbf{x}=\mathbf{x}_0}. \tag{S7}$$

## 3   Background of the von Mises Distribution

### 3.1   Analogy between the von Mises and the normal distribution

A von Mises distribution is defined as

$$\mathcal{M}(x;\mu,\kappa) = \frac{1}{2\pi I_0(\kappa)}\exp\left[\kappa\cos(x-\mu)\right], \tag{S8}$$

with $x$ and $\kappa$ denote the mean and concentration respectively. $I_0(\kappa)$ is the modified Bessel function of first kind and zero order. When $\kappa$ is large, we let $\xi = \kappa^{1/2}(x-\mu)$, and the von Mises distribution is approximated to be

$$\mathcal{M}(\xi;0,\kappa) \propto \exp\left(-\kappa[1-\cos(\kappa^{-1/2}\xi)]\right). \tag{S9}$$

Further approximating $1 - \cos(\kappa^{-1/2}\xi) = \frac{1}{2}\kappa^{-1}\xi^2 + \mathcal{O}(\kappa^{-2})$ for small $\xi$, we have

$$\mathcal{M}(\xi;0,\kappa) \propto \exp\left(-\xi^2/2\right) \propto \mathcal{N}(\xi;0,1). \tag{S10}$$

Thus, the von Mises distribution can be approximated to be a normal distribution for large $\kappa$ and small $|x-\mu|$, i.e,

$$\mathcal{M}(x;\mu,\kappa) \approx \mathcal{N}(x;\mu,\kappa^{-1}). \tag{S11}$$

## 3.2 Product of two von Mises distributions

The posterior of $s_{\text{int}}$ under integration model $m_{\text{int}}$ is the product of two von Mises distributions (see Eq. 19 in the main text)

$$
\begin{aligned}
p(s_{\text{int}}|\mathcal{D}, m_{\text{int}}) &\propto \mathcal{M}(s_{\text{int}}|x_1, \kappa_1)\mathcal{M}(s_{\text{int}}|x_2, \kappa_2), \\
&\propto \exp\left[\kappa_1 \cos(s_{\text{int}} - x_1) + \kappa_2 \cos(s_{\text{int}} - x_2)\right]
\end{aligned}
\tag{S12}
$$

For the terms inside exponential function in above equation, we have,

$$
\begin{aligned}
&\kappa_1 \cos(s_{\text{int}} - x_1) + \kappa_2 \cos(s_{\text{int}} - x_2) \\
&= \kappa_1(\cos x_1 \cos s_{\text{int}} + \sin x_1 \sin s_{\text{int}}) + \kappa_2(\cos x_2 \cos s_{\text{int}} + \sin x_2 \sin s_{\text{int}}), \\
&= (\kappa_1 \cos x_1 + \kappa_2 \cos x_2)\cos s_{\text{int}} + (\kappa_1 \sin x_1 + \kappa_2 \sin x_2)\sin s_{\text{int}}, \\
&= \hat{\kappa}_{\text{int}} \cos(s_{\text{int}} - \hat{s}_{\text{int}}),
\end{aligned}
\tag{S13}
$$

where

$$
\begin{aligned}
\hat{\kappa}_{\text{int}} &= \left[(\kappa_1 \cos x_1 + \kappa_2 \cos x_2)^2 + (\kappa_1 \sin x_1 + \kappa_2 \sin x_2)^2\right]^{1/2}, \\
&= \left[\kappa_1^2 + \kappa_2^2 + 2\kappa_1\kappa_2 \cos(x_1 - x_2)\right]^{1/2};
\end{aligned}
\tag{S14}
$$

$$
\hat{s}_{\text{int}} = \tan^{-1}\left(\frac{\kappa_1 \sin x_1 + \kappa_2 \sin x_2}{\kappa_1 \cos x_1 + \kappa_2 \cos x_2}\right),
\tag{S15}
$$

After normalization, we get

$$
p(s_{\text{int}}|\mathcal{D}, m_{\text{int}}) = \frac{1}{2\pi I_0(\hat{\kappa}_{\text{int}})} \exp\left[\hat{\kappa}_{\text{int}} \cos(s - \hat{s}_{\text{int}})\right].
\tag{S16}
$$

The parameters of above von Mises distributions (Eqs. S14 and S15) can be expressed as complex numbers,

$$
\hat{\kappa}_{\text{int}} e^{j\hat{s}_{\text{int}}} = \kappa_1 e^{jx_1} + \kappa_2 e^{jx_2},
\tag{S17}
$$

where $\kappa e^{jx}$ denotes a vector in polar coordinates, with $\kappa$ and $x$ representing the length and angle of the vector, respectively.

## 4 Neural Encoding Model

We present the mathematical details in deriving the likelihood function of stimulus parameters $\mathbf{w}$ conveyed by neural population code. For the simplicity of notation, we ignore the index of sensory modality $l$ in the derivation below. In the brain, the information of each sensory cue $\mathbf{d}$ from sensory modality is conveyed by the feedforward inputs $\mathbf{u}$ from unisensory areas. Consider the feedforward inputs received by $N$ multisensory neurons are Poisson spikes, $\mathbf{u} = \{u_j\}_{j=1}^N$ (Fig. 1A, top), which satisfies,

$$
p(\mathbf{u}|\boldsymbol{\lambda}) = \prod_{j=1}^N \text{Poisson}(u_j|\lambda_j) = \prod_{j=1}^N \frac{\lambda_j^{u_j}}{u_j!} e^{-\lambda_j},
\tag{S18}
$$

where $\boldsymbol{\lambda}_l$ is the firing rate of $\mathbf{u}_l$. In most experiments, e.g., [1, 2], the measured firing rate $\boldsymbol{\lambda}_l$ is a function of moving direction $s$ and motion coherence $R$, which are the parameters of stimulus presented to animal subjects (Fig. 1A, bottom) [1, 3],

$$
\lambda_j(s, R) = R \exp\left[a \cos(s - \theta_j) - a\right],
\tag{S19}
$$

where $R$ and $a$ determine the peak firing rate and the tuning width respectively. $\theta_j$ is the preferred direction of $j$-th feedforward input, and all inputs' preference $\{\theta_j\}_{j=1}^N$ are considered uniformly cover the space of $s$ and thus forming a *homogeneous* population code.

Substituting Eq. (S19) into Eq. (S18), it could be derived that,

$$
p(\mathbf{u}|s, R) \propto R^{\sum_j u_j} \exp\left[a \sum_j u_j \cos(s - \theta_j)\right] \exp\left[-R \sum_j e^{a \cos(s - \theta_j)}\right].
\tag{S20}
$$

To simplify notations, we denote

$$
\Lambda = \sum_j u_j, \quad \beta = \sum_j e^{a \cos(s - \theta_j) - a}.
\tag{S21}
$$

where $\Lambda$ is the total spike count of feedforward inputs, and $\beta$ is the sum of normalized firing rate of all neurons. In a homogeneous population code, i.e., inputs' preference $\{\theta_j\}_{j=1}^N$ uniformly cover the space of $s$, and $\beta$ is a constant value irrelevant with the amount of input spikes. Working out the sum of the trigonometric function in Eq. (S20) with similar math calculations presented in Sec. 3.2,

$$a \sum_j u_j \cos(s - \theta_j) = \kappa \cos(s - x) \tag{S22}$$

where

$$
\begin{aligned}
x &= \arg\left(\sum_j u_j e^{i\theta_j}\right) = \tan^{-1}\left(\frac{\sum_j u_j \sin\theta_j}{\sum_j u_j \cos\theta_j}\right), \\
\kappa &= a\Lambda\left(\Lambda^{-1} \sum_j u_j \cos\theta_j\right) = a\Lambda\rho(\mathbf{u}).
\end{aligned}
\tag{S23}
$$

$x$ is the moving direction conveyed by feedforward input $\mathbf{u}$ and can be read out by population vector [4], and $\kappa$ quantifies the reliability of the populaiton vector. $\rho(\mathbf{u}) \in [0, 1]$ is the *mean resultant length* of the normalized feedforward inputs [5], and characterizes the width of $\mathbf{u}$. And $\rho(\mathbf{u})$ is irrelevant with amount of spikes, and increases with tuning concentration $a$. It can be derived that the mean of $\rho(\mathbf{u})$ given firing rate $\boldsymbol{\lambda}$ is,

$$
\begin{aligned}
\langle\rho(\mathbf{u})\rangle_{p(\mathbf{u}|\boldsymbol{\lambda})} &= \sum_{j=1}^N \cos\theta_j e^{a\cos\theta_j}, \\
&= NI_1(a),
\end{aligned}
\tag{S24}
$$

where $I_1(a)$ is the modified Bessel function of the first kind and first order.

Using the simplified notations, the likelihood function can be reorganized as,

$$p(\mathbf{u}|s, R) \propto e^{\kappa\cos(s-x)} \times R^\Lambda e^{-\beta R}. \tag{S25}$$

We see $x$, $\kappa$ and $\Lambda$ are linear projections of $\mathbf{u}$, and are *sufficient statistics* in determining the likelihood function of $s$ and $R$ from $\mathbf{u}$. Using the sufficient statistics of $\mathbf{u}$, the likelihood function $p(\mathbf{u}|\boldsymbol{\lambda})$ can be normalized into (Fig. 1B),

$$
\begin{aligned}
p(\mathbf{d} = \{x, \Lambda\}|\mathbf{w} = \{s, R\}) &= p(x|s, \Lambda)p(\Lambda|R), \\
&= \mathcal{M}(x|s, a\rho\Lambda)\,\mathrm{Poisson}(\Lambda|\beta R), \\
&\propto \mathcal{M}(s|x, a\rho\Lambda)\,\Gamma(R|\Lambda + 1, \beta),
\end{aligned}
\tag{S26}
$$

where $\mathcal{M}(x|\mu, \kappa)$ denotes a von Mises distribution over $x$ with mean $\mu$ and concentration $\kappa$, and $\mathrm{Poisson}(x|\lambda)$ is a Poisson distribution over $x$ with rate $\lambda$. $\Gamma(x|\alpha, \beta)$ denotes a Gamma distribution over $x$ with shape parameter $\alpha$ and rate parameter $\beta$. The $x$ and $\Lambda$ correspond to the sensory cue $\mathbf{d}$ in Eq. (1), and the stimulus parameters $s$ and $R$ correspond to $\mathbf{w}$ in Eq. (9).

## 5  Theoretical Calculation of Model Evidence

We present the math details of calculating the model evidence given each model through using Laplacian approximation [6]. We first calculate the model evidence of all observations, i.e., $p(\mathcal{D}|m_h)$, and then work out the model evidence of heading direction, i.e., $p(\mathbf{x}|m_h)$. Recall that the model evidence can be approximated as (Eq. 14) which is defined as

$$
\begin{aligned}
p(\mathcal{D}|m_h) &= \int p(\mathcal{D}|\mathcal{W}_h)p(\mathcal{W}_h|m_h)d\mathcal{W}_h, \\
&\simeq p(\mathcal{D}|\hat{\mathcal{W}}_h)p(\hat{\mathcal{W}}_h|m_h)\det(\mathcal{H}_h/2\pi)^{-\frac{1}{2}},
\end{aligned}
\tag{S27}
$$

To simplify notations, we denote the integrand in above equation as $f(\mathcal{W}_h)$,

$$
\begin{aligned}
f(\mathcal{W}_h) &= p(\mathcal{D}|\mathcal{W}_h)p(\mathcal{W}_h|m_h), \\
&\propto \begin{cases} \prod_{l=1}^2 \mathcal{M}(s_{l,\mathrm{seg}}|x_l, \kappa_l)\Gamma(R_{l,\mathrm{seg}}|\Lambda_l + 1, \beta), & m_{\mathrm{seg}} \\ \prod_{l=1}^2 \mathcal{M}(s_{\mathrm{int}}|x_l, \kappa_l)\Gamma(R_{\mathrm{int}}|\Lambda_l + 1, \beta), & m_{\mathrm{int}}. \end{cases}
\end{aligned}
\tag{S28}
$$

## 5.1 Evidence of the segregation model

Since the prior $p(s_{1,\text{seg}}, s_{2,\text{seg}})$ and $p(R_{1,\text{seg}}, R_{2,\text{seg}})$ are both uniform distribution (Eqs. 3 and 10),

$$f(\mathcal{W}_{\text{seg}}) \quad \propto \quad \prod_{l=1}^{2} \mathcal{M}(s_{l,\text{seg}}|x_l, \kappa_l)\Gamma(R_{l,\text{seg}}|\Lambda_l + 1, \beta). \tag{S29}$$

The logaritham of $f(\mathcal{W}_{\text{seg}})$ is

$$\ln f(\mathcal{W}_{\text{seg}}) = \sum_{l=1}^{2} \kappa_l \cos(s_{l,\text{seg}} - x_l) + \Lambda_l \ln R_{l,\text{seg}} - \beta R_{l,\text{seg}}. \tag{S30}$$

**MAP estimate of stimulus parameters**

Through taking the derivative of $\ln f(\mathcal{W}_{\text{seg}})$ over each parameter to be zero, the maximum-a-posteriori estimate under model $m_{\text{seg}}$ can be found as

$$\begin{aligned}
\frac{\partial \ln f(\mathcal{W}_{\text{seg}})}{\partial s_{l,\text{seg}}} &= -\kappa_l \sin(s_{l,\text{seg}} - x_l) = 0, \quad \Rightarrow \quad \hat{s}_{l,\text{seg}} = x_l, \\
\frac{\partial \ln f(\mathcal{W}_2)}{\partial R_{l,\text{seg}}} &= \frac{\Lambda_l}{R_{l,\text{seg}}} - \beta = 0, \quad \Rightarrow \quad \hat{R}_{l,\text{seg}} = \Lambda_l/\beta.
\end{aligned} \tag{S31}$$

**Occam factor**

Next, we calculate the negative Hessian matrix $\mathcal{H}_{\text{seg}}$. Because the posterior of $s_{l,\text{seg}}$ and $R_{l,\text{seg}}$ ($l = 1, 2$) are conditionally independent with each other given cue inputs $\mathcal{D}$ (Eq. S29), $\mathcal{H}_{\text{seg}}$ is a diagonal matrix. It can be calculated to be,

$$-\frac{\partial^2}{\partial s_{l,\text{seg}}^2} \ln f(\mathcal{W}_{\text{seg}}) \bigg|_{s_{l,\text{seg}}=\hat{s}_{l,\text{seg}}} = \kappa_l, \quad -\frac{\partial^2}{\partial R_{l,\text{seg}}^2} \ln f(\mathcal{W}_{\text{seg}}) \bigg|_{R_{l,\text{seg}}=\hat{R}_{l,\text{seg}}} = \frac{\beta^2}{\Lambda_l}. \tag{S32}$$

And the determinant of $\mathcal{H}_{\text{seg}}$ can be calculated as (using Eq. S23),

$$\det \mathcal{H}_{\text{seg}} = \prod_{l=1}^{2} \kappa_l \frac{\beta^2}{\Lambda_l} = a^2 \rho^2 \beta^4. \tag{S33}$$

The $\det(\mathcal{H}_{\text{seg}})$ doesn't rely on the amount of spikes $\Lambda_l$. This is because the uncertainty (Hessian of log-posterior) of $s_{l,\text{seg}}$ in the posterior decreases with $\Lambda_l$, while the uncertainty of $R_{l,\text{seg}}$ increases with $\Lambda_l$, and their joint effects are completely cancelled with each other. Since the segregation model $m_{\text{seg}}$ has four parameters, i.e., $s_{l,\text{seg}}$ and $R_{l,\text{seg}}$ for $l = 1, 2$, the Occam factor of segregation model is,

$$\begin{aligned}
\text{OF}(m_{\text{seg}}) &= p(\hat{\mathcal{W}}_{\text{seg}}|m_{\text{seg}}) \det(\mathcal{H}_{\text{seg}}/2\pi)^{-\frac{1}{2}}, \\
&= \frac{1}{L_s^2 L_R^2} \sqrt{\frac{(2\pi)^4}{a^2 \rho^2 \beta^4}}, \\
&= \frac{(2\pi)^2}{L_s^2 L_R^2 a \rho \beta^2}.
\end{aligned} \tag{S34}$$

Eventually, the evidence of segregation model can be obtained through substituting Eqs. (S31 and S34) back into Eq. (S27),

$$\begin{aligned}
p(\mathcal{D}|m_{\text{seg}}) &\simeq p(\mathcal{D}|\hat{\mathcal{W}}_{\text{seg}}) \times \text{OF}(m_{\text{seg}}), \\
&\simeq \left[\prod_{l=1}^{2} \mathcal{M}(x_l|x_l, a\rho\Lambda_l) \text{Poisson}(\Lambda_l|\Lambda_l)\right] \frac{(2\pi)^2}{L_s^2 L_R^2 a \rho \beta^2}.
\end{aligned} \tag{S35}$$

## 5.2 Evidence of the integration model

For the integration model, we have,

$$f(\mathcal{W}_{\text{int}}) \propto \prod_{l=1}^{2} \mathcal{M}(s_{\text{int}}|x_l, \kappa_l)\Gamma(R_{\text{int}}|\Lambda_l + 1, \beta). \tag{S36}$$

Taking the logarithm of above equation,

$$\begin{aligned}
\ln f(\mathcal{W}_{\text{int}}) &= \sum_{l=1}^{2}[\kappa_l \cos(s_{\text{int}} - x_l) + \Lambda_l \ln R_{\text{int}} - \beta R_{\text{int}}], \\
&= \sum_{l=1}^{2} \kappa_l \cos(s_{\text{int}} - x_l) + (\Lambda_1 + \Lambda_2)\ln R_{\text{int}} - 2\beta R_{\text{int}}. \tag{S37}
\end{aligned}$$

**MAP estimate of stimulus parameters**

Taking the derivative of $\ln f(\mathcal{W}_{\text{int}})$ over each parameter to be zero, the maximum-a-posteriori estimates of parameter under model $m_{\text{int}}$ are,

$$\begin{aligned}
\frac{\partial \ln f(\mathcal{W}_{\text{int}})}{\partial s_{\text{int}}} &= -\sum_{l=1}^{2} \kappa_l \sin(s_{\text{int}} - x_l) = 0, \quad \Rightarrow \quad \hat{s}_{\text{int}} = \tan^{-1}\left(\frac{\kappa_1 \sin x_1 + \kappa_2 \sin x_2}{\kappa_1 \cos x_1 + \kappa_2 \cos x_2}\right), \\
\frac{\partial \ln f(\mathcal{W}_{\text{int}})}{\partial R_{\text{int}}} &= \frac{\Lambda_1 + \Lambda_2}{R_{\text{int}}} - 2\beta = 0, \Rightarrow \quad \hat{R}_{\text{int}} = \frac{\Lambda_1 + \Lambda_2}{2\beta}. \tag{S38}
\end{aligned}$$

**Occam factor**

The negative Hessian matrix of integration model $m_{\text{int}}$ is a two dimensional diagonal matrix, because the posterior of $s_{\text{int}}$ and $R_{\text{int}}$ are conditionally independent.

$$\begin{aligned}
-\frac{\partial^2}{\partial s_{\text{int}}^2}\ln f(\mathcal{W}_{\text{int}})\bigg|_{s_{\text{int}}=\hat{s}_{\text{int}}} &= \hat{\kappa}_{\text{int}} = \sqrt{\kappa_1^2 + \kappa_2^2 + 2\kappa_1\kappa_2 \cos(x_1 - x_2)}, \\
-\frac{\partial^2}{\partial R_{\text{int}}^2}\ln f(\mathcal{W}_2)\bigg|_{R_{\text{int}}=\hat{R}_{\text{int}}} &= \frac{4\beta^2}{\Lambda_1 + \Lambda_2}. \tag{S39}
\end{aligned}$$

Thus the determinant of $\mathcal{H}_{\text{int}}$ is

$$\det \mathcal{H}_{\text{int}} = \hat{\kappa}_{\text{int}}\frac{\beta^2}{\Lambda_1 + \Lambda_2}.$$

With the assumption that $|x_1 - x_2| \ll \min(\kappa_1, \kappa_2)$, $\hat{\kappa}_{\text{int}} \approx \kappa_1 + \kappa_2$, which will be satisfied when $\Lambda_l$ is large enough,

$$\det \mathcal{H}_{\text{int}} \approx (\kappa_1 + \kappa_2)\frac{\beta^2}{\Lambda_1 + \Lambda_2} = 4a\rho\beta^2. \tag{S40}$$

Combining the above results together, the Occam factor of the integration model is,

$$\begin{aligned}
\text{OF}(m_{\text{int}}) &= p(\hat{\mathcal{W}}_{\text{int}}|m_{\text{int}})\det(\mathcal{H}_{\text{int}}/2\pi)^{-\frac{1}{2}}, \\
&= \frac{1}{L_s L_R}\sqrt{\frac{(2\pi)^2}{4a\rho\beta^2}}, \\
&= \frac{\pi}{L_s L_R \sqrt{a\rho}\beta}. \tag{S41}
\end{aligned}$$

Finally, the evidence of integration model $m_{\text{int}}$ is (substituting Eqs. S38 and S41 into Eq. S27),

$$\begin{aligned}
p(\mathcal{D}|m_{\text{int}}) &\simeq p(\mathcal{D}|\hat{\mathcal{W}}_{\text{int}}) \times \text{OF}(m_{\text{int}}), \\
&\simeq \left[\prod_{l=1}^{2}\mathcal{M}\left(x_l|\hat{s}_{\text{int}}, a\rho\Lambda_l\right)\text{Poisson}\left(\Lambda_l\big|\tfrac{\Lambda_1+\Lambda_2}{2}\right)\right]\frac{\pi}{L_s L_R \sqrt{a\rho}\beta}. \tag{S42}
\end{aligned}$$

Comparing the Occam factors of two models (Eqs. S34 and S41), the $\text{OF}(m_{\text{int}})$ is proportional to the square root of $\text{OF}(m_{\text{seg}})$, and is larger than $\text{OF}(m_{\text{seg}})$ under the parameter setting in current study. This is because the segregation model has more parameters than the integration model, and thus will be penalized more by a smaller Occam factor.

## 5.3  Likelihood of moving direction

We present the math details in approximating the likelihood of moving direction $x$ given stimulus parameters $s$ and $R$ by using Laplace's method, which requires to maginalize the population spike count $\Lambda$,

$$
\begin{aligned}
p(x|s, R) &= \int p(x, \Lambda|s, R)d\Lambda, \\
&= \int \mathcal{M}(x|s, a\rho\Lambda)\text{Poisson}(\Lambda|\beta R)d\Lambda.
\end{aligned}
\tag{S43}
$$

First, we approximate both the von Mises distribution and the Poisson distribution by Gaussian distributions in above equation,

$$
\begin{aligned}
p(x|s, R) &= \int \mathcal{N}\left[x|s, (a\rho\Lambda)^{-1}\right]\mathcal{N}(\Lambda|\beta R, \beta R)d\Lambda, \\
&\approx \mathcal{N}\left[x|s, (a\rho\hat{\Lambda})^{-1}\right]\mathcal{N}[\hat{\Lambda}|\beta R, \beta R]\det(\mathbf{H}/2\pi)^{-1/2},
\end{aligned}
\tag{S44}
$$

where

$$
\hat{\Lambda} = \arg\max_{\Lambda} p(x, \Lambda|s, R), \quad \mathbf{H} = -\left.\frac{\partial^2 \ln p(x, \Lambda|s, R)}{\partial \Lambda^2}\right|_{\Lambda=\hat{\Lambda}}.
\tag{S45}
$$

The calculation of $\hat{\Lambda}$ and $\mathbf{H}$ is presented below. To simplify notations, we use $\mathcal{L}$ to denote the likelihood $p(x, \lambda|s, R)$.

$$
\ln \mathcal{L} = \frac{1}{2}\ln \Lambda - \frac{a\rho\Lambda}{2}(s - x)^2 - \frac{(\Lambda - \beta R)^2}{2\beta R}.
\tag{S46}
$$

$\hat{\Lambda}$ can be found by taking the derivative of $\mathcal{L}$ over $\Lambda$ to be zero,

$$
\begin{aligned}
\frac{\partial \ln \mathcal{L}}{\partial \Lambda} &= \frac{1}{2\Lambda} - \frac{a\rho}{2}(s - x)^2 - \frac{\Lambda - \beta R}{\beta R} = 0, \\
\hat{\Lambda} &= \left[1 - \frac{a\rho}{2}(s - x)^2\right]\frac{\beta R}{2} + \frac{1}{2}\sqrt{\left[1 - \frac{a\rho}{2}(s - x)^2\right]^2\beta^2 R^2 + 2\beta R}, \\
&\approx \left[1 - \frac{a\rho}{2}(s - x)^2\right]\beta R + \mathcal{O}(R^{1/2}).
\end{aligned}
\tag{S47}
$$

To gain theoretical insight, we simplify above equation by omitting the term $2\beta R$ inside the square root function . This approximation works well when $R$ is large enough, since $R^2$ is a order larger than $R$.

On the other hand, the negative Hessian matrix is,

$$
\begin{aligned}
\left.\frac{\partial^2 \mathcal{L}}{\partial \Lambda^2}\right|_{\Lambda=\hat{\Lambda}} &= -\frac{1}{2\hat{\Lambda}^2} - \frac{1}{R}, \\
&= -\frac{1}{2[1 - \frac{a\rho}{2}(s - x)^2]^2\beta^2 R^2} - \frac{1}{R}, \\
&\approx -\frac{1}{R} + \mathcal{O}(R^{-2}).
\end{aligned}
\tag{S48}
$$

This approximation is also considered under the large $R$ limit where the omitted term is a order smaller than $1/R$.

Substituting Eqs. (S47 and S48) back into Eq. (S44), we get an unnormalized likelihood for moving direction $x$,

$$
p(x|s, R) \propto \sqrt{\hat{\Lambda}(x)}\exp\left[-\frac{a\rho\hat{\Lambda}(x)}{2}(s - x)^2\right]\exp\left[-\frac{(\hat{\Lambda}(x) - \beta R)^2}{2\beta R}\right].
\tag{S49}
$$

Since this distribution is complicated, again, we approximate it by a Gaussian distribution in order to simplify our theoretical analysis. It is easy to see $p(x|s, R)$ is a symmetric distribution over its center $x = s$, and it is a mixture of Gaussian distributions with different width. Next, we find the Hessian of $p(x|s, R)$ at it peak location, $x = s$. Similarly, denote by $\mathcal{L}(x) = p(x|s, R)$ to simplify notations, and we have,

$$\ln \mathcal{L}(x) = \frac{1}{2} \ln \hat{\Lambda}(x) - \frac{a\rho \hat{\Lambda}(x)}{2}(s - x)^2 - \frac{(\hat{\Lambda}(x) - \beta R)^2}{2\beta R}. \tag{S50}$$

The Hessian of $\mathcal{L}(x)$ is calculated as

$$\frac{\partial \ln \mathcal{L}(x)}{\partial x} = \frac{1}{2\hat{\Lambda}(x)}\frac{\partial \hat{\Lambda}(x)}{\partial x} - \frac{a\rho(s-x)^2}{2}\frac{\partial \hat{\Lambda}(x)}{\partial x} + a\rho \hat{\Lambda}(x)(s-x) - \frac{\hat{\Lambda}(x) - \beta R}{\beta R}\frac{\partial \hat{\Lambda}(x)}{\partial x},$$

$$\frac{\partial^2 \ln \mathcal{L}(x)}{\partial x^2} = -\frac{1}{2\hat{\Lambda}(x)^2}\left(\frac{\partial \hat{\Lambda}(x)}{\partial x}\right)^2 + \left[\frac{1}{2\hat{\Lambda}(x)} - \frac{a\rho(s-x)^2}{2} - \frac{\hat{\Lambda}(x)}{\beta R} + 1\right]\frac{\partial^2 \hat{\Lambda}(x)}{\partial x^2}$$

$$+ \left[2a\rho(s-x) - \frac{1}{\beta R}\right]\frac{\partial \hat{\Lambda}(x)}{\partial x} - a\rho \hat{\Lambda}(x).$$

Meanwhile, the derivative of $\hat{\Lambda}(x)$ over $x$ is

$$\frac{\partial \hat{\Lambda}(x)}{\partial x} = a\rho\beta R(s - x), \quad \frac{\partial^2 \hat{\Lambda}(x)}{\partial x^2} = -a\rho\beta R.$$

And then we have,

$$\frac{\partial^2 \ln \mathcal{L}(x)}{\partial x^2} = -\frac{[a\rho(s-x)]^2}{2[1 - \frac{a\rho}{2}(s-x)^2]^2} - \frac{a\rho}{2[1 - \frac{a\rho}{2}(s-x)^2]} + \frac{5}{2}a^2\rho^2\beta R(s-x)^2 - a\rho(s-x) - a\rho\beta R.$$

And the Hessian of $\mathcal{L}(x)$ at $x = s$ is,

$$\frac{\partial^2 \ln \mathcal{L}(x)}{\partial x^2}\bigg|_{x=s} = -a\rho\left(\frac{1}{2} + \beta R\right),$$

$$\approx -a\rho\beta R. \tag{S51}$$

We throw out $1/2$ in above equation since it is much smaller than $\beta R$ when $R$ is large. Finally, the likelihood can be approximated as a Gaussian distribution, or a von Mises distribution (using the analogy between the Gaussian and von Mises distribution in SI. 3.1),

$$p(x|s, R) = \mathcal{N}\left[x|s, (a\rho\beta R)^{-1}\right],$$

$$\approx \mathcal{M}(x|s, a\rho\beta R). \tag{S52}$$

Fig. S1 suggests this approximation works very well under the parameters considered in our study.

# 6 Neural Implementation of Integration and Bayes Factor

We present the math details of how the population of neurons could compute and represent the posterior of heading direction in the integration model, i.e., $p(s_{\text{int}}|\mathcal{D}, m_{\text{int}})$, and the likelihood ratio of heading direction of cue 1, i.e., $\text{LR}(x_1)$.

## 6.1 Implementation of integration

From Eq. (S20), the unnormalized likelihood function of heading direction $s_{\text{int}}$ in the integration model given feedforward input $\mathbf{u}_l$ is,

$$p(s_{\text{int}}|\mathbf{u}_l) \propto \exp\left[a\sum_j u_l(\theta_j)\cos(s_{\text{int}} - \theta_j)\right], \tag{S53}$$

where $u_l(\theta_j)$ denotes the feedforward input from modality $l$ with preferred direction $\theta_j$ given cue $l$. Then the posterior of $s_{\text{int}}$ is proportional to,

$$p(s_{\text{int}}|\mathbf{u}_1, \mathbf{u}_2) \propto p(\mathbf{u}_1|s_{\text{int}})p(\mathbf{u}_2|s_{\text{int}}),$$

$$\propto \exp\left[a\sum_j (u_1(\theta_j) + u_2(\theta_j))\cos(s_{\text{int}} - \theta_j)\right]. \tag{S54}$$

Suppose the posterior $p(s_{\text{int}}|\mathbf{u}_1, \mathbf{u}_2)$ is represented by a population of $N$ multisensory neurons $\mathbf{r}^c$, whose tuning is also satisfied by Eq. (S19), and thus the posterior decoded from the population activity $\mathbf{r}^c$ is

$$p(s_{\text{int}}|\mathbf{r}^c) \propto \exp\left[a\sum_j r^c(j)\cos(s_{\text{int}} - \theta_j)\right]. \tag{S55}$$

Equating the terms inside the exponential functions in above two equation,

$$\sum_j r^c(j)\cos(s_{\text{int}} - \theta_j) = \sum_j u_1(\theta_j)\cos(s_{\text{int}} - \theta_j) + \sum_j u_2(\theta_j)\cos(s_{\text{int}} - \theta_j). \tag{S56}$$

Equating the coefficients of every cosine terms, we have,

$$r_1^c(j) = u_1(\theta_j) + u_2(\theta_j). \tag{S57}$$

It suggests that the posterior of heading direction after integration could be represented by a population of neurons whose responses are the sum of two feedforward inputs together. This is consistent with a previous study [7].

Moreover, the preferred direction of $r^c(j)$ under two cues are both $\theta_j$, and thus this is consistent with the congruent neuron discovered in MSTd and VIP [1, 2].

## 6.2 Implementation of Bayes factor

Suppose there is another population of neurons $\mathbf{r}^c$ representing the likelihood ratio $\text{LR}(x_1)$. Similar with the derivations of implementing the integration in Sec. 6.1, from Eqs. (24-25), the neuronal responses should satisfy,

$$\sum_j r^o(j)\cos(x_1 - \theta_j) = \frac{1}{2}\left[\sum_j u_1(\theta_j)\cos(x_1 - \theta_j) - \sum_j u_2(\theta_j)\cos(x_1 - \theta_j)\right], \tag{S58}$$

Above equation indicate that the Bayes factor might be implemented by

$$r^o(j) = \frac{1}{2}[u_1(\theta_j) - u_2(\theta_j)]. \tag{S59}$$

However, since the firing rate $r^o$ is only a positive number, the responses $r^o$ will be rectified when $u_2(\theta_j)$ is larger than $u_1(\theta_j)$. And then there would be rectification error.

This problem could be solved through using the property of cosine functions,

$$\begin{aligned}
\sum_j r^o(j)\cos(x_1 - \theta_j) &= \frac{1}{2}\left[\sum_j u_1(\theta_j)\cos(x_1 - \theta_j) + \sum_j u_2(\theta_j)\cos[x_1 - (\theta_j + \pi)]\right], \\
&= \frac{1}{2}\left[\sum_j u_1(\theta_j)\cos(x_1 - \theta_j) + \sum_j u_2(\theta_j + \pi)\cos(x_1 - \theta_j)\right], \\
&= \frac{1}{2}\left[\sum_j [u_1(\theta_j) + u_2(\theta_j + \pi)]\cos(x_1 - \theta_j)\right], \tag{S60}
\end{aligned}$$

Equating the coefficients of the same cosine terms in above equation, the neuronal response in representing Bayes factor should satisfy,

$$r^o(j) = \frac{1}{2}\left[u_1(\theta_{1j}) + u_2(\theta_j + \pi)\right]. \tag{S61}$$

It means the neuronal response $\mathbf{r}^o$ is the sum of $\mathbf{u}_1$ from modality 1 and the inputs $\mathbf{u}_2$ from modality 2 but being rotated to the opposite direction. Therefore, the preferred direction of $r^o(j)$ over stimulus of sensory modality 1 is $\theta_j$, while it becomes $\theta_j + \pi$ under sensory modality 2. This is consistent with the tuning of opposite neurons obsrved in experiments [1, 2].

## 7 Simulation Details and Model Parameters

To confirm the validity of the neural implementation for integration and Bayes factor, we conducted a simulation consisting a population of congruent neurons and another population of opposite neurons. And then compare the decoded quantity from congruent neurons and opposite neurons with the theoretical predictions of integration (Eq. 20) and likelihood ratio in Bayes factor (Eq. 25) respectively.

## 7.1 Simulation details and model parameters

We list the typical parameters of our model. The width of space of moving direction $L_s = 2\pi$. And the width of space of tuning curve peak $R$ is $L_R = 100$, consistent with that a cortical neurons' firing rate can be up to about 100Hz. And the tuning concentration $a = 3$, corresponding to the tuning width of approximately $40°$. There are $N = 180$ input neurons for each sensory modality.

In the simulation performed in Fig. 3E-F, we consider a time window $\Delta t = \beta^{-1}$, which makes the average spike counts $\langle \Lambda_l \rangle$ of feedforward input $\mathbf{u}_l$ is $R\beta\Delta t = R$. The direction of stimulus 1, $s_1$ is fixed at $0°$, while the direction of stimulus 2, $s_2$ is ranging from $0°$ to $60°$ with a step of $10°$. Moreover, the strength of each stimulus $R_l$ ranges from 5Hz to 50Hz with a step of 5Hz independently. Given a combination of stimulus parameters, the firing rates of two feedforward inputs $\boldsymbol{\lambda}_1$ and $\boldsymbol{\lambda}_2$ are calculated (Eq. 7), and then we use the Poisson spike generator to produce Poisson spikes $\mathbf{u}_1$ and $\mathbf{u}_2$ as feedforward inputs (Eq. 6). For each combination of stimulus parameters, it was repeated for 50 trials. Within each trial, the feedforward spiking inputs are generated.

### Neural responses and decoding

The responses of congruent neurons implementing integration are obtained by summing the two spiking inputs together, i.e., $\mathbf{r}^c = \mathbf{u}_1 + \mathbf{u}_2$ in Eq. (26). In contrast, the responses of "opposite" neurons implementing the likelihood ratio of cue 1, $\text{LR}(x_1)$, will be obtained through combining the two inputs in an opposite way, i.e., $\mathbf{r}^o(j) = [\mathbf{u}_1(\theta_j) + \mathbf{u}_2(\theta_j + \pi)]/2$ in Eq. (27).

Once the neuronal responses are obtained in a trial, we decode the mean and concentration of the posterior of direction encoded by congruent neurons,

$$
\begin{aligned}
\hat{s}(\mathbf{r}^c) &= \arg\left(\sum_j r_j^c e^{i\theta_j}\right) = \tan^{-1}\left(\frac{\sum_j r_j^c \sin\theta_j}{\sum_j r_j^c \cos\theta_j}\right), \\
\hat{\kappa}(\mathbf{r}^c) &= a\sum_j r_j^c \cos\theta_j.
\end{aligned}
\tag{S62}
$$

Then $\hat{s}(\mathbf{r}^c)$ and $\hat{\kappa}(\mathbf{r}^c)$ decoded from congruent neurons will be compared with the theoretical predictions $\hat{\kappa}_{\text{int}}$ and $\hat{s}_{\text{int}}$ obtained from Eq. 20. The result is plotted in Fig. 3E.

On the other hande, we decode the mean and concentration from opposite neurons by using the same way as Eq. (S62), which are denoted by $\hat{s}(\mathbf{r}^o)$ and $\hat{\kappa}(\mathbf{r}^o)$ respectively. $\hat{s}(\mathbf{r}^o)$ and $\hat{\kappa}(\mathbf{r}^o)$ are supposed to the estimate of $x_{1p}$ and $\kappa_{1p}$ (Eq. 24) computed by opposite neurons. Meanwhile, the parameters of $\text{LR}(x_2)$ will be obtained by $x_{2p} = \hat{s}(\mathbf{r}^o) + \pi$, and $\kappa_{2p} = \hat{\kappa}(\mathbf{r}^o)$. Then we substitute the estimates of $x_{1p}$, $x_{2p}$, $\kappa_{1p}$ and $\kappa_{2p}$ into Eqs. (23-24) to calculate the Bayes factor read out by opposite neurons. Finally, the results will be compared with the Bayes factor computed by theoretical prediction, and are plotted in Fig. 3F.

### Theoretical prediction

For each feedforward input $\mathbf{u}_l$, we decode $x_l$ and $\kappa_l$ from it by using population vector as specified in Eq. (8). And then we substitute the decoded $x_1$, $x_2$, $\kappa_1$ and $\kappa_2$ into Eq. (20) to calculate $\hat{s}_{\text{int}}$ and $\hat{\kappa}_{\text{int}}$, which will be used as the theoretical prediction of the posterior of heading direction after integration.

On the other hand, we substitute the decoded $x_1$, $x_2$, $\Lambda_1$ and $\Lambda_2$ into Eq. (25) to get the parameters of likelihood ratio, i.e., $x_{lp}$ and $\kappa_{lp}$, and then calculate the value of likelihood ratio $\text{LR}(x_1)$ and $\text{LR}(x_2)$ (Eq. 24). Finally, the Bayes factor could be calculated by using Eq. (23).