[Reviews · NeurIPS 2019]

Reviewer 1



After reading the Author Feedback: I thank the authors for the useful explanations and comments, in particular with respect to novelty and comparison to related work, which I encourage them to include in the revised paper. Overall, I think this is an interesting contribution but still somewhat niche (e.g., it applies only to circular variables), and somewhat incremental with respect to recent similar work; and empirical evidence is only suggestive. Nevertheless, there are enough novel theoretical contributions to potentially deserve publication in NeurIPS; so I confirm my score. Summary: This paper proposes how neurons could implement perceptual causal inference (that is, decide between integration or segregation of two sensory cues from distinct modalities, such as vision and vestibular sense) via population coding and simple linear operations, at least for cues living in a circular space (e.g., heading direction). The major difficulty here is the computation of the Bayes factor so as to decide which model (integration or segregation) best describes the data. The authors show that by assuming that the brain infers stimulus reliability as well as stimulus direction, and via some approximations (e.g., Laplace), all the computations become tractable. Interestingly, the theory requires that in order to compute the Bayes factor, the brain should have neurons and anti-neurons tuned to opposite directions (from different modalities) which act simultaneously; which is compatible with experimental findings. Originality: Medium. The proposal for computing Bayes factors is novel as far as I know, but there is some overlap with recent work which is not fully addressed in the paper (see detailed comments). However, the mathematical techniques used here could lead to further developments of population codes. Quality: High. The problem is interesting, and the analysis and quality of the work is high throughout. Clarity: The paper is well-structured; the text is well-written and the mathematical content well-explained, and the images provide useful information. Significance: This is a potentially interesting contribution as it would potentially explain some experimental findings -- on the other hand, there seems to be considerable overlap with previous literature, which would make this paper still technically interesting for how causal inference is implemented, but somewhat incremental. Major comments: This is a generally thorough and well-written paper, with a considerable amount of additional material to expand upon the main text. My main concern with the work is that the authors should better clarify in the manuscript what their original contributions are, in particular with respect to prior work: - Zhang et al. (2016), which is cited in passing in the current submission, but not expanded upon; - Most importantly, Zhang et al. (2018), which is not cited in the current submission, and with which there seems to be a certain amount of overlap. The same work seems also to have been recently published in eLife (Zhang et al., 2019), but note that the preprint has been available online at least since November 2018. Second, it is not completely clear from the paper why the authors' proposal succeeds where previous work has failed (in particular, reference [15]). The key point, also stressed by the authors, seems to be that, in this submission, the stimulus reliability R is treated explicitly as part of the inference (see lines 99-106), whereas it "was treated as a 'nuisance' parameter in the previous studies". However, being a nuisance parameter merely means that it gets marginalized away, and this study too eventually has to perform a marginalization (Eq. 14), although on Lambda (the *observed* spike count), which is likely the key difference. Can the authors elaborate on why their approach is successful (as opposed to the previous ones)? In short, it seems that the paper would strongly benefit from an extended "Related work" section which highlights the novelty and strengths of the current submission. Minor comments: line 39: Note that the Bayes factor is the ratio of the *posteriors* over models -- which is equivalent to the marginal likelihood ratio for an equal prior over models, but this (while being a common choice) is still a specific case. Figure 1: Panel A uses a non-standard depiction for latent vs. observed variables in graphical models. A fairly standard depiction in statistics and machine learning is to use "filled gray" nodes for observed variables, and solid-circle (white) nodes for latent variables (although if the authors are familiar with another convention, that's fine too). Besides this negligible point, the figure is very clear and well-made. line 71: w_int is a parameter vector, not a single parameter, and it includes both s and R. A uniform distribution over s is non-problematic, since it is a naturally bounded dimension (uniform over the circle), but I reckon that you assume a *bounded* uniform distribution over R (and not an improper prior); as explained later (Eq. 10). Perhaps here just add that you use a "bounded uniform" distribution, to avoid ambiguity. I am stressing this apparently innocuous detail because the choice of bounds (more in general, the choice of prior) is *crucial* for the computation of the marginal likelihood; an arbitrary large interval for R can arbitrarily penalize the more complex model. Typos: The paper is generally clear and very well-written, but there is a bunch of typos here and there, such as occasionally missing articles ("the"), or redundant ones. Given the otherwise high quality of the writing, I recommend to double-check the paper and supplementary information for spelling and grammar. I am pointing out a few of these errors here, but there is likely much more to fix. line 15: amendable --> amenable line 24: "often are hierarchy with latent variables" --> unclear phrasing line 75: "the prior of two models" --> "the prior of the two models" line 83: function over --> function of line 84: the width of tuning function --> the width of the tuning function line 85: the reliability of stimulus --> the stimulus reliability line 137: "In the below, we presented how" --> "In the section below, we present how" line 161: "of two models" --> "of the two models" line 166: "In above equation" --> "In the above equation" line 171: "When the cues are from the same object, their consensus is similar with themselves statistically" --> rephrase better line 175: "normalizing constant" --> "normalizing constants" line 183: "two red vectors" --> "the two red vectors" line 184: "two likelihood ratios" --> "the two likelihood ratios" line 185: "the opposite means" --> "opposite means" line 187: "irrelevant with" --> "irrelevant of" line 191: "two cues’ consensus to explain cues" --> "the two cues’ consensus to explain the cues" line 191: "explaining power" --> "explanatory power" line 218: "opposite with" --> "opposite to" line 219: "reveals" --> "reveal" line 227: "stimlus" --> "stimulus" line 231: "with the equal number" --> "with equal number" line 246: "addresses causal inference" --> "addresses how causal inference" Supplementary Material: line 30: Anglogy --> Analogy line 33: with first kind --> of first kind line 90: logritham --> logarithm line 92, 110 (and perhaps others): maximum-a-posterior --> maximum-a-posteriori line 108: Taking logarithm --> Taking the logarithm line 118: Combining above results --> Combining the above results line 126: requires maginalize --> requires to maginalize line 131: "is presented in the below" --> "is presented below" line 134: through omitting --> by omitting line 144: "it is a mixture of Gaussian" --> it is not a mixture, it's the product of two distributions line 181: consine --> cosine line 183: should satisfies --> should satisfy References: Zhang, W. H., Wang, H., Wong, K. M., & Wu, S. (2016). “Congruent” and “opposite” neurons: sisters for multisensory integration and segregation. In Advances in Neural Information Processing Systems (pp. 3180-3188). Zhang, W. H., Wang, H., Chen, A., Gu, Y., Lee, T. S., Wong, K. M., & Wu, S. (2018). Concurrent Multisensory Integration and Segregation with Complementary Congruent and Opposite Neurons. bioRxiv, 471490. Zhang, W. H., Wang, H., Chen, A., Gu, Y., Lee, T. S., Wong, K. M., & Wu, S. (2019). Complementary congruent and opposite neurons achieve concurrent multisensory integration and segregation. eLife, 8, e43753.

Reviewer 2



This paper proposes a Bayesian model for how neural circuits decide whether observed stimulus features arose from a common stimulus (i.e., under a cue integration model) or whether observations arose from unrelated stimuli (under a cue segregation model). The idea is that a neural circuit can perform model selection by computing Bayes factors, which favour an integration model when the disparity between cues is small. The computation of Bayes factors is implemented by two populations of neurons: a population of "congruent" neurons and a population of "opposite" neurons. Such populations have been observed experimentally, and the model provides a parsimonious explanation of the experimental data. The main contribution of the paper is a derivation of the Bayes factor for a population of Poisson-spiking neurons using a number of analytically convenient assumptions and approximations, and a computational model illustrating how a component of the Bayes factor can be implemented in neural networks. The modelling work is interesting and to a high standard, and I believe provides novel insight (both analytic and geometric) into the conditions under which integration vs segregation is favoured. The authors suggest that their work provides the first rigorous formulation of how the opposite neurons could play a central role in the encoding of Bayes factors. One drawback of the work, as the authors acknowledge, is that they only provide a network model that calculates one component of the Bayes factor, and it is unclear how the remaining components (the Occam factors) could be calculated by the network. I found that this point was not sufficiently addressed until the final sentence of the discussion. This is at odds with the abstract, which seems to imply that the authors had a complete implementation. The authors could be more transparent about this. Also, at times the writing was rather unclear, occasionally with words seemingly mistakenly omitted (e.g. on line 35); the paper could benefit from further proof reading.

Reviewer 3



The authors describe a way to compute Bayes factors that could be implemented by neurons to carry out model/causal inference. My primary concern with this paper is that while the authors describe a plausible inference rule for neuronal computation, they do not actually present any empirical evidence to support their claim. The authors argue that the proposed inference method may be implemented by "opposite neurons". While I do agree that "opposite neurons" support the plausibility of the authors' arguments, I feel that for a NeurIPS submission stronger evidence would be warranted. As such, I think this is a decent paper with some interesting ideas, but that is not sufficiently comprehensive for NeurIPS. On a side note, the authors appear to equate Bayes factor computation with causal inference. I would recommend a more careful wording here. In my opinion, computing Bayes factors may -- but does not need to be -- one step in a causal inference algorithm. Update after rebuttal/discussion: The authors and other reviewers have convinced me that there is value in this work without strong empirical evidence. I have raised my score from 4 to 6.

[Author Response · NeurIPS 2019]

We appreciate the constructive and detailed review comments suggested by three reviewers. We will revise all typos and proofread the manuscript carefully in a revised version. Followings are our reply to three reviewers' major concerns.

**1. Comparison with related works.** Earlier works on the computation of opposite neurons (e.g., Zhang et al., eLife 2019; bioRxiv 2018; NIPS 2016) only suggested that opposite neurons compute the likelihood ratio. Here, we derived a mathematically rigorous link using a normative theory, and explained opposite cells' functions in the theoretical framework of causal inference. This we believe is a fundamental, rather than incremental, advance in theoretical neuroscience. In detail, the math derivations of opposite neurons in previous works started directly from a likelihood ratio defined similarly as Eq. (24) in the current paper, which are now grounded on the novel generative model and causal inference as presented in Eqs. (1-23) in the current paper. Secondly, the distribution ratios computed by opposite neurons have distinct physical meanings between current work and earlier works. Now, the integration and segregation models reconstruct their respective best-fit likelihoods over the input directions $x$ (Eq. 16), which are represented by the half of the blue vector (from the origin to the centroid of the parallelogram) and green vectors in Fig. 2B respectively. And opposite neurons compute the ratio between the best-fit likelihoods reconstructed from two models, which are geometrically the two red vectors emanating from the centroid of the parallelogram to either green vector. In contrast, in previous works, opposite neurons compute the posterior ratio representing the disparity of the latent stimulus direction $s$ inferred from each of two cues respectively ($\overline{\text{Eq. 5 in}}$ Zhang 2019), which is geometrically the difference between two green vectors (Fig. 3B in Zhang 2019). Lastly, since the length of the vector representation of the likelihood ratio computed by opposite neurons in current work (Fig. 2B) is half of the one in previous works, in the implementation, opposite neurons average two inputs in our work (Eq. 27), while they sum two inputs together in previous works (e.g., last Eq. on page $\overline{19 \text{ in Zhang}}$ 2019). Taken together, this paper is significant in that we fully developed and established the theoretical link of opposite neurons to causal inference, and illuminate the functional roles of opposite neurons.

As pointed out by reviewer 1, ref. [15] was not able to compute Bayes factor by a linear population code, and we are able to do by including the stimulus strength $R$ in the generative model which leads to two effects. First, the constant Occam factor with respect to inputs simplifies the network implementation (see Sec. 2). Second, the reliability of the best-fit likelihood in the integration model $m_{\text{int}}$ ($\hat{\kappa}_{\text{int}}/2$ in Eq. 24) is the average of the reliability of two likelihoods ($\hat{\kappa}_{\text{int}} \approx \kappa_1 + \kappa_2$ in Eq. 20), because the estimate $\hat{R}_{\text{int}}$ is the average of all input spike counts (Eq. 21). In the implementation, this implies the best-fit likelihood of the model $m_{\text{int}}$ can be represented by the average of all population inputs, which can be realized by linear operation. On the other hand, without the stimulus strength $R$ as in ref. [15], the reliability of the likelihood ratio ($\kappa_{lp}$ in Eq. 24) and the Occam factor ratio $\text{OF}(m_{\text{seg}})/\text{OF}(m_{\text{int}})$ both depend on inputs nonlinearly, hence it is impossible to use linear operations to compute the Bayes factor. Moreover, in a generative model without $R$, the best-fit likelihood of the model $m_{\text{int}}$ is $\mathcal{M}(x_l|\hat{s}_{l,\text{int}}, \kappa_l)$, which has the same reliability but different means with the likelihood. This requires a neural circuit to move the location of population inputs (representing the likelihood) in stimulus feature space while keeping the spike count unchanged, which is hard to implement.

**2. The implementation of Occam factor.** We will discuss the potential implementation of the Occam factor calculation in a biologically plausible neural circuit model in the revised paper. Eq. (22) and line 161-162 states that the Occam factor is a constant invariant to the sufficient statistics of inputs, i.e., $x_l$ and $\Lambda_l$ ($l = 1, 2$), which indicates the Occam factor ratio can be represented by a fixed parameter in the network, simplifying the network implementation significantly. For example, we can consider a scenario that a downstream neuron whose firing probability represents the logarithm of Bayes factor, i.e., $\ln \mathcal{B}(\mathbf{x}) = \sum_{l=1}^{2} \ln \text{LR}(x_l) + \ln[\text{OF}(m_{\text{seg}})/\text{OF}(m_{\text{int}})]$. The downstream neuron receives the inputs $\ln \text{LR}(x_l)$ from opposite neurons, and receives a constant background input or has a firing threshold representing the Occam factor ratio $\ln[\text{OF}(m_{\text{seg}})/\text{OF}(m_{\text{int}})]$.

**3. The significance of our work.** How neural circuits solve perceptual causal inference is an important and open question in neuroscience (also see reviewer 1's comments). The key contribution of our work is in establishing a theoretical link between the "empirical evidence" of opposite cells found in multi-sensory areas (Fig. 3C-D) and the theoretical framework of causal inference. Our work provides novel insights to the functional roles of opposite cells in causal inference. Thus, the neurophysiological observation of the opposite cells is the "empirical evidence" for the biological plausibility of the Bayes factor computation in neural circuit. We provide the theoretical derivations and simulations to demonstrate how Bayes factor could be computed in opposite neurons, thus linking the normative theory and the underlying neural substrates. More importantly, our theory can provide predictions for verifying the proposed functions of opposite neurons in future experiments. For example, our hypothesis suggests that activating or inactivating opposite neurons' activities would decrease or increase the behavioral choice of integration model (or integration probability) in perception. We will make this prediction more explicit in the Discussion in a revised manuscript.

Related works on causal inference in the field (see references [10, 13, 16]) used generative models with similar structure as our study. Thus, the terminology we used follows the earlier works in the field. We'd like to revise the wording if necessary. We admit that Bayesian model selection (Bayes factor) might not be the only way to solve causal inference, though the existence of the opposite neurons provides strong evidence in supporting it.

[Meta-Review · NeurIPS 2019]

This is a borderline paper and was discussed extensively by the reviewers after reading the authors' rebuttal. The reviewers agree that the method and analysis for the Bayes factor computation is novel. However, the relation of the paper to prior published work is not elucidated and there is concern that it may be a little niche in its appeal.